# Unique Benefits of Tumor-Specific Nanobodies for Fluorescence Guided Surgery

**DOI:** 10.3390/biom11020311

**Published:** 2021-02-18

**Authors:** Thinzar M. Lwin, Robert M. Hoffman, Michael Bouvet

**Affiliations:** 1Department of Surgery, University of California San Diego, San Diego, CA 92093, USA; helmi.lwin@gmail.com (T.M.L.); all@anticancer.com (R.M.H.); 2Department of Surgery, Dana Farber Cancer Center, Boston, MA 02215, USA; 3Department of Surgery, VA San Diego Health Care System, San Diego, CA 92161, USA; 4AntiCancer Inc., San Diego, CA 92111, USA

**Keywords:** fluorescent nanobody, tumor-specific fluorescence, fluorescence-guided surgery

## Abstract

Tumor-specific fluorescence labeling is promising for real-time visualization of solid malignancies during surgery. There are a number of technologies to confer tumor-specific fluorescence. Antibodies have traditionally been used due to their versatility in modifications; however, their large size hampers efficient fluorophore delivery. Nanobodies are a novel class of molecules, derived from camelid heavy-chain only antibodies, that have shown promise for tumor-specific fluorescence labeling. Nanobodies are ten times smaller than standard antibodies, while maintaining antigen-binding capacity and have advantageous features, including rapidity of tumor labeling, that are reviewed in the present report. The present report reviews special considerations needed in developing nanobody probes, the status of current literature on the use of nanobody probes in fluorescence guided surgery, and potential challenges to be addressed for clinical translation.

## 1. Introduction

Cancer surgery requires real-time, in-situ localization of the lesion and its safe and complete removal. The goal is an R0 resection, which results in no gross or microscopic tumor remaining at the primary tumor bed. Current approaches rely on cross sectional imaging with knowledge of adjacent anatomy along with visual inspection, palpation, and experience. Cross sectional imaging is usually obtained weeks in advance of surgery and does not provide real time feedback once the anatomy is distorted by surgical mobilization. Intraoperative imaging modalities such as ultrasound, CT or MRI are either limited in their contrast modalities or are bulky, costly, and impractical for routine use. Intra-operative tumor frozen sections are limited in their scale/scope and are subject to sampling error as it is impossible to sample the entire tumor bed to ensure complete resection.

A majority of oncologic resections are performed under bright light visualization using the naked eye. The reliance on subjective information leads to significant variability in estimating the completeness of resection, reflected by the wide variability in rates of margin positive resections [1]. Despite the experience and meticulous inspection by the surgeon, positive margins lead to rapid locoregional recurrence and poor overall outcomes [2,3,4,5]. This indicates a need for a more precise technique to enhance the ability of surgeons to assess tumors in real-time [6]. However, there have been limited advances in the surgical approach to improve rates of cure after resection.

The use of fluorescence has emerged as a novel technology to aid intra-operative visualization of tumors in relation to anatomic structures [7]. Fluorescence-guided surgery (FGS) is a contact-free approach which uses a camera equipped with specialized filters to capture a signal emitted by endogenous or exogenously-administered fluorescence from the tumor after excitation with an appropriate light source. This is an ideal modality which incorporates well with the surgical workflow as many surgical procedures today are already performed under image-guidance using white light, i.e., laparoscopic and robotic surgeries.

Contrast enhancement of the tumor with fluorescence enables the surgeon to better delineate and view its anatomic relationship in-situ and in real time with immediate feedback as the tissue is being manipulated. Pre-clinical studies have shown that the use of FGS is promising in improving tumor resection and oncologic outcomes. Metildil et al. established proof of principle in resection of expressing green fluorescent protein (GFP) tagged colon cancer tumors in orthotopic mouse models under bright light surgery (BLS) versus fluorescence guidance. Using FGS increased rates of complete resection (100% FGS vs. 58% BLS), decreased rates of recurrence (33% FGS vs. 62% BLS), and increased disease-free survival (Median >36 weeks FGS vs. 9 weeks BLS). The median overall survival increased (31 weeks FGS vs. 16 weeks BLS) as well as rates of cure as defined by “alive without evidence of tumor at >6 months post-surgery” (67% FGS vs. 37% BLS) [8]. The improved oncologic outcomes using FGS was similar in other cancers such as pancreatic cancer in orthotopic mouse models [9].

Fluorescent dyes such as fluorescein, methylene blue, 5-aminolevulinic acid and indocyanine green (ICG) have been useful for evaluating perfusion of anatomic structures and or mapping the pattern of lymph node basins [10,11,12,13]. The enhanced permeability and retention (EPR) effect of tumors allows selective tumor retention of macromolecules, especially lipophilic ones [14,15]. It is thought that the molecules extravasate from leaky tumor vessels with disorganized venous and lymphatic drainage and remain retained in solid tumors. This principle has been exploited for visualization of pancreatic cancer, glioblastomas and a number of thoracic malignancies using the lipophilic dye, ICG [16,17,18]. A high dose of the dye is administered several days prior to imaging to allow slow accumulation in the tumors. However the fluorescence accumulation from using the dye alone can be heterogenous without the ability to specifically direct molecule to the tumor. Conjugation of fluorophores to an active targeting moiety addresses this issue.

## 2. Platforms for Tumor-Specific Probes

An optimal probe for tumor-specific delivery of fluorescence would specifically label the neoplastic lesion with a high contrast compared to adjacent normal tissue and identify the presence or absence of affected lymph nodes, or distant disease [7]. The agent should be safe, cost-effective to produce, have good molecular stability, have rapid pharmacokinetics, and a high sensitivity and specificity to the target [19]. The body of literature accumulated in identifying markers unique to cancer cells or their microenvironment has been applied to logical probe design to produce a number of tumor-targeting fluorescent agents.

Antibodies, with their inherent ability for recognition and binding of epitopes have been widely studied for tumor-specific delivery of conjugated molecules such as drugs or radio-tracers [20]. An antibody targeting the desired antigen can be conjugated to near-infrared dyes to create a tumor-specific probe for FGS. Pre-clinical studies have shown that the use of fluorescently labeled anti-carcino-embryonic antigen (CEA) antibodies for FGS led to an improvement in complete resection rates (85.7% BLS vs. 95.5% FGS) [21]. Even when there is residual tumor, FGS decreases the volume of residual tumor [22].

A number of clinical trials are currently ongoing to evaluate the efficacy of antibody fluorophore conjugates in visualizing cancers during surgery [23]. This is well summarized in the review by Hernot et al. Table 1 [23]. Clinical trials that have had the greatest progress have targeted vascular-endothelial growth factor (VEGF), epidermal growth factor receptor (EGFR), and CEA. Fluorescent anti-VEGF and EGFR antibodies are currently undergoing phase I and II clinical trials [24,25,26,27]. Fluorescent anti-CEA antibody (SGM-101) is currently undergoing a phase III clinical trial [28,29]. The studies have shown that there is clear visualization in situ within the surgical field as well as ex vivo in the excised lesion.

The large size of the antibody molecule can lead to delayed and limited penetration at the tumor. Additionally, Fc receptor-mediated recycling of antibodies lead to a long serum half-life which is not desirable for an imaging agent. There is an overall delay in peak fluorescence accumulation at the tumor and washout of the unbound probe. The delay can be up to several days between injection and optimal imaging. Attempts to decrease the size of antibodies (IgG = 150 kDa) to smaller antigen-binding fragments such as minibodies [38], scFv’s [39,40,41], diabodies [42] and Fab [43,44] fragments for FGS were evaluated in pancreatic cancer, colon cancer, breast cancer, prostate cancer, and head and neck cancer. However, these antibody fragments were unstable and difficult to optimize for high yield production of the fluorophore conjugates [45]. There was a higher contrast obtained at earlier time points compared to intact antibodies, but at a decreased total overall tumor signal.

The discovery of a subset of IgG-like antibodies in camelids consisting of only heavy chains, led to the isolation of a single-domain antibody (sdAb) [46]. These monomers are only 12–15 kDa in size and they retain a similar affinity and specificity compared to classical IgG’s [47]. They are the smallest naturally occurring antigen binding units and are called nanobodies.

Nanobodies are being actively evaluated for tumor-specific drug delivery and tumor-specific nuclear imaging [48]. They are starting to be explored for tumor-specific fluorophore delivery [49]. Nanobodies have uniquely advantageous properties as a platform for molecular targeting. Nanobodies have high thermal and chemical stability which permit unique conjugation strategies in harsh conditions that would otherwise denature the three-dimensional conformation of a classical antibody [50,51]. Nanobodies are hydrophilic, highly soluble, and have limited association with other hydrophobic protein surfaces [52]. Despite their monovalent structure, they demonstrate target binding affinity at nanomolar concentrations [53]. Due to their unique shape, they can penetrate epitopes deep within cryptic clefts [54,55]. They are cost-effective to produce in bacteria or yeast with a high yield [56].

Due to their high homology with the human variable heavy chain fragment, nanobodies have limited immunogenicity. Clinical trials using nanobodies as either therapeutic agents or as a radio-tracer have shown no evidence of immunogenicity [57,58,59,60]. The molecules were produced in E. Coli without lipopolysaccharide contamination and the reported endotoxin level was found to be only 0.01 EU/mg [61]. Caplacizumab, an anti-von Willebrand factor (vWF) nanobody, developed for the treatment of thrombotic thrombocytopenic purpura (TTP) is the most developed nanobody, having completed phase III clinical trials [59,62]. Anti-Her2 nanobodies conjugated to radiotracers (I-131 and Ga-68) to image patients with breast cancer have completed phase I clinical trials and are undergoing phase II clinical trials [58,63]. A summary of nanobodies currently under clinical trials is well summarized in Jovčevska et al. Table 1 [51].

## 3. Rational Probe Design for FGS with Nanobodies

Nanobodies have been conjugated successfully to fluorophores, a majority in the NIR range, for optical imaging. Traditionally, antibodies are conjugated to fluorophores by creating an amide bond between primary amines on antibodies and a fluorophore bearing an activated N-hydroxysuccinimide (NHS) ester group [64]. The approach is simple, rapid, has a high conjugate yield, and can be readily applied to nanobodies. However, NHS conjugation can lead to variabilities in the number and location of fluorophores and a heterogenous product in both antibodies and nanobodies.

Efforts have been made to direct the site of conjugation as the heterogeneity has been shown to impact the safety, pharmacokinetics and efficacy of the molecule [65]. This impact is greater in very small molecules such as nanobodies. For example, the random placement of fluorophores in nanobodies has led to obstruction of the antigen binding site and decreased avidity of the probe [23]. This results in an overall decreased contrast at the region of interest and increased off target signal. Therefore, site-directed labeling of fluorophores is indicated for small molecule probes and cysteine-malemide conjugation has been commonly used in developing nanobody-NIR probes for FGS [66]. Cysteine-malemide conjugation involves the introduction of a cysteine residue on the surface of the molecule to provide a reactive sulfhydryl groups for conjugation [67,68]. In nanobody conjugation, the cysteine is usually placed at the carboxyl terminus, therefore positioning the conjugation site on the opposite side of the antigen binding region [66]. Besides cysteine-malemide chemistry, alternative conjugation strategies such as trans-glutaminase, sortase, or azido-alkyne click chemistry can be explored in the future due to high chemical stability of nanobodies [69,70,71].

Choice of fluorophores can also affect bioavailability and pharmacokinetics of the probe as the charge of the fluorophore impacts the net charge of the molecule [72]. Common NIR fluorophores and their status in clinical use are well summarized in Hong et al. Table 1 [73]. Visible wavelength fluorophores and near-infrared fluorophores have been used with targeting probes in clinical studies, but near-infrared fluorophores (680 nm or higher) are advantageous due to their improved tissue penetration, decreased auto-fluorescence and light scattering [74]. Most commercially available NIR fluorophores are hydrophobic. ICG carries a charge of −1 and IRDye800CW is −4 [75]. Both have significant hepatobiliary excretion and GI tract accumulation with a high off-target background signal [72]. Targeting probes in clinical trials have been conjugated to IRDye800CW since the dye is readily available and has been demonstrated to show safety and efficacy in Phase I/II clinical trials [33,34]. Anti-VEGF antibodies (bevacizumab-800CW [33]) and anti-EGFR antibodies tagged to IRDye800CW (panitumumab-IRDye800CW and cetuximab-IRDye800CW [34]) are such examples. Additionally, devices to detect NIR fluorescence at the 800 nm range are already in place in modern operating rooms for detection of ICG perfusion assessment. Spectral overlap between ICG and IRDye 800CW avoids the need to develop or purchase additional imaging devices for tumor-specific fluorescence imaging. Neutral or zwitterion-based fluorophores, in Phase I clinical trials, have the potential to preferentially direct the overall probe to decrease serum binding, lower non-specific tissue accumulation, and increase renal elimination, while maintaining a bright fluorescence signal at the tumor with a high extinction coefficient and quantum yield [76,77,78]. Zwitterion conjugated nanobodies can be explored for future use.

The number of fluorophores a molecule carries impacts the overall performance of the molecule [79]. The optimal number of fluorophores a molecule should carry has not yet been determined. While it is ideal for a probe to carry as many fluorophores as possible without affecting tumor targeting, densely packed fluorophores in proximity lead to self-quenching [80]. This results in significantly lower than expected fluorescence intensities for the given number of fluorophores [81]. A high degree of labeling, similar to random labeling can lead to obstruction and decreased tumor binding [82]. Due to their small size, it is anticipated that nanobodies will carry fewer fluorophores than antibodies, but further research in this area is needed.

## 4. Current Initiatives in Fluorescent Nanobody Probes for FGS

A summary of nanobody probes being developed for tumor-specific FGS, is shown in Table 1. Nanobodies described in current literature have been conjugated to fluorophores using either NHS or cysteine-maleimide chemistry. Fluorescence signals at the tumor were detectable in less than an hour. Tumor-to-background (TBR) values ranged from 2–3 in-vivo and 6.6–42 ex-vivo [22,23]. The highest TBR was commonly obtained at about 2–4 h when using site-directed conjugation approaches and 24 h when using NHS conjugation [19,20]. A variety of enclosed and open animal imagers as well as clinical fluorescence imaging devices were used to detect the fluorescence signal.

Nanobodies against epidermal growth factor receptor (EGFR), a trans-membrane receptor overexpressed in many epithelial cancers were first explored for FGS. Oliviera et al. developed 7D12, an anti-EGFR nanobody conjugated to IRDye800CW using NHS chemistry in a subcutaneous epidermoid carcinoma mouse model in 2012 [19]. The work was the first to compare the anti-EGFR antibody, cetuximab and the anti-EGFR nanobody 7D12 side by side for the purpose of optical imaging and tumor-specific FGS. They showed that 7D12-IR800 demonstrated tumor specific fluorescence as early as 30 min after injection and imaging with the IVIS Lumina (Perkin Elmer, Waltham, MA). In comparison, there was no signal above background at the very early time point of 30 min with cetuximab-IR800. However, after 4 h, the signals between the antibody and nanobody probes were comparable, with the antibody probe showing a delayed clearance. They showed that there was up to 17% of the injected dose per gram of tumor (ID/g) 2 h after injection of the nanobody while there was up to 10% ID/g after injection of cetuximab. It is interesting to note that although the authors comment on the time point of 2 h post-injection as yielding the clearest signal with the nanobody probe, the tumor-to background ratio (TBR) at that time point was about 1.5 and a higher TBR of approximately 2.3 was reached at 24 h. TBR for cetuximab-IR800 at 24 h was around 2 and remained above 2 for up to 72 h. The clearest image using cetuximab-IR800 was at 24 h, consistent with the literature in antibody fluorophore conjugates.

7D12-IR800 showed efficacy in an orthotopic mouse model of squamous cell head and neck cancer, even detecting cervical lymph node metastases in-situ [20]. The probe was specific for the tumor and co-registered with the GFP expressed by the tumor. A similar TBR of 2.7 at 24 h was found as in the experiment above. The efficacy of the probe was established using the FLARE (Fluorescence-Assisted Resection and Exploration) [83], an open imaging system designed to mimic a surgical environment with greater surrounding light interference compared to the IVIS which is an enclosed box, not suitable for interactive surgical navigation. Sufficient contrast was observed using the FLARE with TBR’s greater than 2 at 24 h. This peak TBR at a 24 h time point with 7D12-IR800 is unique compared to other nanobody probes. Site-specifically labeled probes usually show a peak TBR usually around 2–4 h and level off thereafter.

The human epidermal growth factor receptor 2 (HER2/neu) is also overexpressed in breast, ovarian, and some gastric, lung, and head and neck cancers. Nanobodies targeting HER2 have been evaluated for use in tumor-specific FGS of breast and ovarian cancer. Kijanka et al. developed 11A4, an anti-HER2 nanobody conjugated to IRDye800CW in a site-specific manner, using cysteine-maleimide chemistry [21]. They compared the fluorophore conjugated anti-HER2 antibody, trastuzumab, to 11A4-IR800 in subcutaneous breast cancer mouse models and again showed that nanobodies have an advantage in timing of tumor labeling and contrast. 11A4-IR800 had a TBR of 2.5 versus trastuzumab-IR800 1.4 at 4 h after injection and imaging with the IVIS (Perkin Elmer, Waltham, MA). Interestingly, they showed that trastuzumab-IR800 had accumulation in both HER2 positive and HER2 negative xenografts, while 11A4-IR800 had accumulation only in HER2 positive breast cancer xenografts and no accumulation in HER2 negative tumors. These results indicate that fluorescence signal from trastuzumab-IR800 at HER2 negative tumors may be due to the EPR effect of tumors rather than antibody antigen binding. When comparing the site-specific nanobody to randomly labeled nanobodies, site-specific probes had a two-fold higher TBR of approximately 2 after 4 h compared to randomly labeled nanobodies with a TBR of approximately 1.

Debie et al. also targeted HER2 with the site-specific fluorophore conjugated nanobody 2Rs15dCys-IRDye800 in a mouse model of intraperitoneal ovarian cancer [22]. This model produces many tumor implants of variable sizes on organs and peritoneal tissue. The probe was effective for detection of even sub-millimeter lesions when imaged with the FluoBeam800 (Fluoptics, Grenoble, France). This was confirmed by co-localization of bioluminescence in luciferase tagged ovarian cancer cells. Sensitivity of detection of peritoneal tumors improved from 59.3% using bright light imaging (BLI) to 99.0% using fluorescence guidance with 2Rs15dCys-IRDye800. The percentage of false positive lesions also decreased from 19.6 to 7.1%. The use of fluorescence guidance was shown to decrease the amount of residual tumor implants after surgery: The percentage of residual tumor implants after traditional BLS was 2.87% vs. FGS 1% vs. BLS followed by FGS 0.71%. The combination of probe and imaging system produced a very high TBR, which ranged from 14.4–42 shown by imaging at 1.5 h after probe injection. This is unusual as other nanobody probes were able to achieve TBR in the 2–4 range. Such high TBR values may be due to the removal of the kidneys before imaging, thereby decreasing the background. Alternatively, the Fluobeam could be a more sensitive device at detection of the fluorescence signal; other work by Debie et al. [23] using the Fluobeam (Table 1) also report TBR values between 6 and 12. This is promising, as the Fluobeam LX is a similar device manufactured by Fluoptics, currently available as a hand-held NIR camera for clinical use [84].

Another target over expressed in solid GI malignancies is the human carcinoembryonic antigen (CEA), a cell surface glycoprotein, commonly used as a serum marker for colorectal and pancreatic cancers [85]. Our laboratory has previously demonstrated the efficacy of anti-CEA antibodies for tumor labeling and surgical navigation [86,87,88,89]. We have now evaluated the efficacy of an anti-CEA nanobody (NbCEA5) site-specifically conjugated to IRDye800 for tumor-specific FGS in an orthotopic mouse model of pancreatic cancer [24]. A fluorescence signal was detectable at the tumor as early as 30 min after injection and imaging with the CRI Maestro (Cambridge Research & Instrumentation, Inc. Woburn, MA) and the Pearl Trilogy (Perkin Elmer, Waltham, MA). The fluorescence signal co-registered with GFP tagged pancreatic cancer cells. A peak TBR of 2.32 was obtained at 2 h. The probe and fluorescence signal were specific to CEA expression, as CEA expressing BxPC-3 pancreatic tumors showed a strong signal with a TBR of 2.66 compared to low CEA expressing MiaPACA2 pancreatic tumors, which had a TBR of only 0.5. This is due to the very low level of CEA expressed in the MiaPACA2 cell line. Even tumors as small as 3 mm that were barely palpable within the pancreatic parenchyma of the mouse were detectable using the fluorescent anti-CEA nanobody. The fluorescence signal was detectable with several clinically available 800 nm NIR imaging devices using the robotic da Vinci Firefly camera (Intuitive, Sunnyvale, CA, USA) and the Stryker AIM laparoscope (Stryker Corporation, Kalamazoo, MI) (Figure 1). Phase III clinical trials evaluating FGS for colorectal malignancies using SGM-101, an anti-CEA antibody conjugated to a 700 nm fluorophore are currently under way (NCT03659448) and an anti-CEA-based nanobody probe is promising.

Rather than targeting molecules overexpressed by tumors, another approach is to detect molecular environments unique to tumors. Hypoxia is present in a majority of solid tumors and absent in healthy tissue. Proteins such as carbonic anhydrase IX (CAIX) are upregulated under hypoxic conditions. CAIX has been shown to be one of the most tumor-specific membrane bound proteins expressed in hypoxic tumors [90]. Van Brussel et al. evaluated CAIX as a target for tumor-specific FGS using anti-CAIX nanobody (B9) site-specifically conjugated to IRDye800 in mouse models of breast cancer [25]. They used the SurgOptix real-time intra-operative multispectral fluorescence reflectance imaging (MFRI) system (SurgOptix, Redwood shores, CA) and found an optimal TBR of 4.3 two hours after injection. The probe was able to detect pre-cancerous ductal-carcinoma in-situ (DCIS) tumors with a TBR of 1.8. A combination of fluorescent nanobodies to both CAIX and HER2 produced an increase in TBR at 2–4 h in an orthotopic breast cancer mouse model [26]. The probes localized to different areas of the tumor based on differential expression of their targets resulting in a higher overall TBR at the tumor. Further work will show if a combination of probes rather than a single target is more efficacious for improved tumor localization.

As the field of nanobody-based probes expands for tumor-specific FGS, additional targets or cocktail of targets, alternative conjugation approaches and optimization strategies will be investigated to further improve intraoperative imaging.

## 5. Potential Challenges, Solutions, and Future Directions

Nanobodies are small molecules and can realistically bear only one fluorophore conjugate per nanobody to avoid issues with blockage of the binding pocket or the close proximity of fluorophores next to each other, leading to self-quenching [80]. It is anticipated that more nanobodies must accumulate at the tumor compared to antibodies to produce an equivalent signal that can be detected by current intra-operative fluorescence imaging devices. Studies in nanobodies have shown that injection of a higher dose of nanobodies does not necessarily lead to a similar increase in background signal as is the case with antibodies [91]. Fluorophore and dose optimization for nanobody-based fluorescence probes will need to be further refined.

Imaging devices will need to be further optimized to maximize sensitivity of detection. While existing mouse studies have shown the benefit of fluorescently conjugated nanobodies as tumor targeting agents, the efficacy of fluorescent nanobody probes must be studied in clinically available devices to evaluate the number of fluorophores as well as the amount of nanobody that must realistically accumulate at a tumor to produce a clinically relevant signal. It is beyond the scope of this review to discuss the details of imaging technology, but there are a number of factors that can be modified to improve the resulting image [92]. The choice of light source, filters, camera capture optics, software processing, registration, camera housing, and display all facilitate the use of fluorescence technology for oncologic resections [93,94]. Current FGS imaging systems in the operating room have a wide range of sensitivities for signal detection with a variety of perform capabilities for display.

While nanobodies can bind deep pockets with nanomolar affinities, the rapid renal elimination of nanobodies leads to a very short serum half-life and a smaller window for exposure and binding to antigen. This can lead to concerns over non-optimal imaging should there be any delays in surgery. However in the pre-clinical studies cited in the present review that measured the fluorescence signal over time, that there is still tumor-specific fluorescence at up to 72 h with TBR’s above 1.5 [19]. As the molecules have rapid accumulation, there is also the potential for repeat dosing of the probe although there remains work to be done in this area to optimize this strategy.

Renal elimination of the probe due to the small size of the molecule can also lead to off target signal. As this is a known issue, nanobody-based molecules cannot be used for kidney tumors. There may also be limitations on tumors with anatomic locations adjacent to the kidney, the kidney may need to be mobilized to avoid shine through of the signal.

Maximal probe delivery still need to be optimized with nanobodies as there are so small with rapid renal elimination. This may not necessarily be a priority with intra-operative imaging agents compared to therapeutic agents given that there is a sufficiently bright enough signal for contrast for the duration of the surgery. However if desired, a potential approach to increase probe delivery is to create multimeric nanobodies which could potentially increase affinities (up to 10–600 fold) [52]. Other approaches include the addition of albumin molecules to decrease renal filtration [95]. Other synthetic molecules such as *N*-(2-hydroxypropyl) methacrylamide (HPMA), a hydrophilic polymer can increase circulation time [96]. This has been shown to be efficacious in peptide-based probes against EGFR, which face similar challenges as nanobodies in clearance [97,98,99]. The goal is titration of serum half-life to extend the window of time for probe and tumor binding for improved tumor labeling. The increased size of additional molecules will lead to slower to tissue penetration and decreased depth of penetration compared to monomeric nanobodies or nanobodies alone [100]. Depth of penetration may not be an issue if the desired result is the tumor to normal tissue interface for margin identification, but the timing to imaging can lead to undesired delays. The optimal balance between an increased time to permit probe binding without causing an increase in background signal is still to be determined.

Depth of signal penetration with optical imaging can be an issue. NIR fluorophore signals have an increased depth of penetration compared to visible wavelength fluorophores, but this is still limited only to 5–7 mm. This issue can potentially be addressed with dual nuclear and optical imaging tracers. Nanobodies can be labeled with PET isotopes with short half-lives such as ^68^Ga or ^18^F. Proof of concept of a dual nuclear and optical tracer was shown using a cRGD peptide-based probe carrying both a ^89^Zr PET tracer and a ZW800 fluorophore in colon cancer animal models [101]. Portable gamma probes for intraoperative localization of tumor deposits are being developed [102]. Nanobody-based nuclear tracers are already in clinical trials and similarly designed molecules with a handheld gamma probe could grossly localize the lesion while fluorescence optical imaging can further delineate the tumor with great precision.

Nanobody technology is an excellent platform that holds promise not just for tumor-specific imaging, but also for tumor-specific therapeutics [103]. Thernostics combines both diagnostic and therapeutics into one concept. The nanobody targeting moiety can carry a fluorophore that is also a photosensitizer such as IRDye700Dx [104]. The fluorophore can be used for fluorescence intra-operative imaging, followed by a therapeutic treatment of the tumor bed to induce production of photo-reactive oxygen species (ROS) which will selectively induce cell death in microscopic deposits of residual tumor cells [105]. This has been shown to be feasible in antibodies bearing photosensitizers and is beginning to be explored in nanobodies bearing photosensitizers. Van Driel et al. showed that photodynamic therapy one hour after IV administration of EGFR-nanobodies conjugated to IRDye700Dx can cause tumor necrosis (approx. 90%) and little to no toxicity in surrounding normal tissues in a mouse model of squamous cell carcinoma of the tongue [106]. Beside photosensitizers, hybrid nanobodies can be created by conjugation with radiolabels for tumor-specific PET imaging as well as therapeutic radionuclides [107]. It is possible to consider the use of one molecule for both pre-operative tumor specific imaging and active intra-operative navigation. Nanobodies can also carry therapeutic drugs or pro-drugs as well as nanoparticles [52,108]. However due to renal filtration of small molecules such a nanobodies, renal toxicity can be a potential hurdle. Nanobody technology can be a possible avenue to combine tumor-specific optical fluorescence imaging with drugs or radionuclides for a multi-modal approach to treat cancer.

## 6. Anticipated Benefits

FGS is a highly promising approach to assist surgeons in visualizing the tumor during surgery with the goal of achieving improved oncologic outcomes. The use of nanobody probes has multiple advantages: The most promising feature is the decreased time to tumor labeling. This has been demonstrated in mouse models described above, as peak TBR is usually obtained within 2–6 h. While the details of translation of pharmacokinetics from murine models to humans may need to be further evaluated, the time frame is promising. Caplacizumab, the anti-vWF nanobody, was shown to have a maximal serum concentration of 20% 6–7 h after subcutaneous administration [60]. This shorter window to optimal imaging is promising for clinical translation as patients are not required to make a separate clinical visit 1–2 days before surgery for probe administration and washout. Fluorescent nanobodies can potentially be administered on the day of the procedure.

Other advantages such as high affinity, stability of the molecule, cost-effective production with high yield in an easily modifiable biologic platforms are additional positive considerations in clinical translation of nanobodies. Initiatives for alternative conjugation chemistry in the field of fluorophore-nanobody conjugates are forthcoming.

Nanobodies are versatile molecules that can be conjugated with NIR fluorescence dyes with promising pharmacokinetics for FGS. They also hold potential for targeting with other molecules such as photosensitizers, nuclear tracers, therapeutic radionuclides, or drugs for multi-modal tumor-specific theranostics.

## 7. Conclusions

Nanobodies offer a promising technology for tumor-specific delivery of fluorescence. The use of a nanobody platform offers advantages in the speed of fluorescence signal delivery, stability of molecule, and versatility in conjugation as well as production. Efforts to optimize dose and avidity to produce an optimal image are underway. All these advantages will increase the effectiveness and extend the use of FGS, a critical advancement in the field of surgical oncology.

## Figures and Tables

**Figure 1 biomolecules-11-00311-f001:**
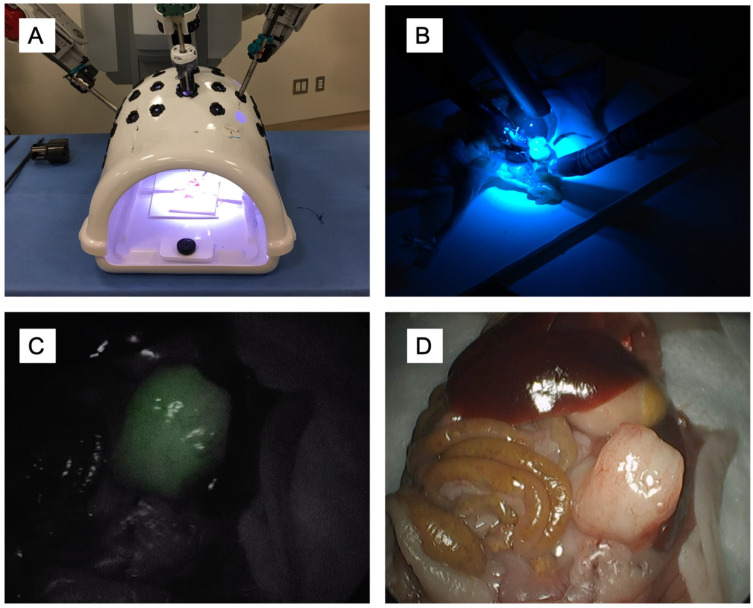
Fluorescence signal is detectable at the tumor using an anti-CEA nanobody conjugated to IRDye800CW using clinically available FDA-cleared fluorescence imaging devices. Images were acquired after placing the animal within the minimally-invasive simulation dome (**A**) and switching between the bright light and the fluorescence mode (**B**). Representative images from fluorescence imaging using the Stryker Aim laparoscopic camera (Kalamazoo, MI, USA) and the corresponding bright light image (**C**,**D**). Tumors are approximately 7 mm–1 cm at the time of imaging.

**Table 1 biomolecules-11-00311-t001:** A summary of pre-clinical nanobody probes being developed for tumor-specific FGS.

Target	Molecule Name	Fluorophore	Nb Conjugation Approach	Dose nb Administered	Time to Peak nb TBR	Peak nb TBR	Imaging Device	Disease
EGFR	7D12 [30]	IRDye 800CW	NHS	25 ug	24 h	2.3′s	IVIS Lumina	Epidermoid carcinoma
	7D12 [31]	IRDye 800CW, IRDye680RD	NHS	75 ug	24 h	2, 2.72 at 24 h	FLARE and IVIS spectrum	Head and neck cancer
Her 2	11A4 [32]	IRDye 800CW	Site directed cysteine-maleimide and randomly labeled	50 ug	4 h	2.5–3	IVIS Spectrum	Breast cancer
	2Rs15d [33]	IRDye 800CW	Site directed cysteine-maleimide	2 nmol	3 h	14.4–42	Fluobeam800	Ovarian cancer
	2Rs15d [34]	IRDye 800CW, IRDye680RD	Site directed cysteine-maleimide and randomly labeled	2 nmol	3 h	6.6 at 3 h, 12 at 24 h	Fluobeam800, FMT2500 Perkin Elmer	Ovarian cancer, breast cancer
CEA	NbCEA5 [35]	IRDye 800CW	Site directed cysteine-maleimide	2 nmol	2 h	2.6	MaestroCRI, Firefly, StrykerAIM	Pancreatic cancer
CAIX	B9 [36]	IRDye800CW	Site directed cysteine-maleimide	50 ug	2–4 h	4.3	SurgOptix	Breast cancer, pre-invasive breast cancer
CAIX & Her2	B9, 11A4 [37]	IRDye800CW, IRDye680RD or IRDye700DX	Site directed cysteine-maleimide	50 ug each of each probe, 100 ug per mouse	2–6 h	2–3.3	IVIS Spectrum	Breast cancer

## Data Availability

Not applicable.

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
