# Peer review of "Unique Benefits of Tumor-Specific Nanobodies for Fluorescence Guided Surgery"

_biomolecules, 2021, doi:10.3390/biom11020311_

Round 1

Reviewer 1 Report

see the review attached

Author Response

February 1, 2021

Dr. Vladimir N. Uversky, PhD, DSc

Editor-in-Chief, Biomolecules

Dear Dr. Uversky and reviewers,

Thank you very much for your careful consideration of our manuscript for publication in Biomolecules. We respect the feedback from the reviewers on this paper and appreciate the opportunity to strengthen specific aspects of this manuscript. In our revised manuscript, we have addressed the comments made by the reviewers point by point, and believe that our responses will satisfy their concerns.

All changes to the document are highlighted in yellow on the revised manuscript. Point-by-point responses (in italics) are given below, with references to changes in the manuscript.

Reviewer 1:

The presented review deals with a topic of a potential interest to wide pool of readers in the field of cancer research, nanomedicine and drug delivery. The authors summarized numerous papers dealing with application of nanobodies as targeting ligands for fluorescently guided surgery (FGS) of malignant tumors.

In the present form, the review is a list of many nanobody conjugates with fluorescent dyes; however, a more detailed discussion of the often-mentioned site-specific synthesis is omitted. The conjugates are categorized mostly according to the target tumor antigen, which is good; unfortunately, the comparison with analogous full-antibody-antibody based probes is sometimes little bit biased in favor of nanobodies though the pharmacokinetic properties of the system antibody are often very similar.

We have edited the section on discussion of conjugation strategies to reflect this suggestion, please see details below.

An important feature of the tumor tissue is increased vascular permeability resulting in EPR effect. Though the EPR effect is once mentioned in the paper, it is not correctly explained (see point 2). EPR effect plays an important role in passive tumor targeting of high-molecular weight compounds; consequently, it should be discussed also in terms of delivery of fluorescent probes to solid tumors.

We have edited the section on discussion of the EPR effect to reflect this suggestion, please see details below.

I recommend publishing of the manuscript in Biomolecules after a major revision reflecting both the above-mentioned general comments and the following specific points:

  1. P. 1, l. 24: What does R0 resection mean?

We have edited the statement to better define R0: ”Cancer surgeries require real-time, in-situ, localization of the lesion and its safe and complete R0 resection in which no gross or microscopic tumor remains in the primary tumor bed.”

  1. P. 2, l. 53: EPR effect is responsible for tumor accumulation of only high-molecular weight compounds tumor.

We have edited this to reflect that EPR accumulation is most distinct in macromolecules and lipophilic molecules. ”The enhanced permeability and retention (EPR) effect of tumors allows selective tumor retention of macromolecules, especially lipophilic ones12,13. It is thought that the molecules extravaste from leaky tumor vessels with disorganized venous and lymphatic drainage and this remain retained at solid tumors. This principle has been exploited for visualization of pancreatic cancer, glioblastomas and a number of thoracic malignancies using the lipophilic dye, indocyanine green (ICG)14–16. A high dose of the dye is administered several days prior to imaging to allow slow accumulation at the lesions”

  1. P. 3, l. 79: Remove the second ”to avoid”.

This has been removed: ”A near infrared dye is desirable to avoid the nonspecific tissue autofluorescence”

  1. P. 3, l. 83: ”…targeted VEGF receptor”

”the” has been removed: ” targeted vascular-endothelial growth factor (VEGF),”

  1. P. 4, l. 94: Explain abbreviation scFv.

We have edited this statement to define scFV’s as ”single-chain variable fragments (scFv’s)”

  1. P. 4, l. 106: A typing error ”They have…”

This has been corrected in the manuscript: ” They have uniquely advantageous properties”

  1. P. 4, l. 142: Which probes? Please, re-write.

This has been editred to further clarify: ” Targeting probes in clinical trials have been conjugated to IRDye800CW since the dye is readily available and have been demonstrated to show safety and efficacy59,60. Anti-VEGF antibodies (bevacizumab-800CW59) and anti-EGFR antibodies tagged to IRDye800CW (panitumumab-IRDye800CW and cetuximab-IRDye800CW60) are such examples.”

  1. P. 5, l. 181: was no signal above background 181 at 30 minutes with cetuximab-This statement is not true. There was an almost comparable signal at the time point 2 h with cetuximab-IR800 as with the 7D12-IR800. The only difference was a bit slower clearance of the signal for the full antibody conjugate after 24 h.

In Olivera et al, the authors state that it was at the very early time point at 30 minutes that there was no signal above background using cetuximab-IR800 compared to the nanobody (Figure 3A). At two hours after injection, the signal is more comparable and after four hours the error bars overlap. This paragraph has been edited to reflect this point. ” The clearest image using cetuximab-IR800 was at 24 hours, consistent with the literature in antibody fluorophore conjugates. “They showed that 7D12-IR800 demonstrated tumor specific fluorescence as early as 30 minutes after injection and imaging with the IVIS Lumina (Perkin Elmer, Waltham, MA). In comparison, there was no signal above background at the very early time point of 30 minutes with cetuximab-IR800. However, after 4 hours, the signals between the antibody and nanobody probes are comparable, with the antibody probe showing a delayed clearance. It is interesting to note that although the authors comment on the time point of 2 hours post-injection as yielding the clearest signal with the nanobody probe, the tumor-to background ratio (TBR) at that time point was about 1.5 and a higher TBR of approximately 2.3 was reached at 24 hours. TBR for cetuximab-IR800 at 24 hours was around 2’s and remained above 2 for up to 72 hours. The clearest image using cetuximab-IR800 was at 24 hours, consistent with the literature in antibody fluor-ophore conjugates.”

  1. P. 5, l. 186: What was the TBR value for the full antibody conjugate cetuximab-IR800 at 24h? This important information is missing…

The authors do not clearly state the value but the TBR based on Figure 3C for cetuximab-IR800 at 24 hours is around 2 and remains above 2’ for the remainder of the time points measured during their experiment. This has been added to the manuscript. ” TBR for cetuximab-IR800 at 24 hours was around 2’s and remained above 2 for up to 72 hours”

  1. P. 6, l. 202: Kijanka et al reference is missing.

This has been corrected in the manuscript.

  1. P. 6, l. 203: Please, specify the site-specific manner of the conjugation reaction. This is the key point of the whole synthesis. The site-specific nanobody conjugation is repeatedly mentioned but never explained in more detail.

The cystiene-maldemide conjugation process is further described in the manuscript. ” Cystiene-malemide conjugation involves the introduction of a cysteine residue on the surface of the molecule to provide a reactive sulfhydryl groups for conjugation53,54. In nanobody conjugation, the cysteine is usually placed at the carboxyl terminus, therefore positioning the conjugation site on the opposite side of the antigen binding region52.”

  1. P. 6, l. 227: An explanation of such a high TBR (compared to the previous systems targeted against the same receptor) is not provided. At least some hypothesis should be offered...

We have suggested some hypothesis for the high TBR and the manuscript has been modified to reflect this. ”This is unusual as other nanobody probes are able to achieve TBR’s in the 2-4 ranges. Such high TBR values may be due to the fact that kidneys which can contribute to high background signal were removed prior to fluorescence, thereby decreasing the background. Alternatively, the Fluobeam could be a more sensitive device at detection of the fluorescence signal; other work by Debie et al21 using the Fluobeam (Table 1) also report TBR values between 6-12’s.”

  1. P. 6, l. 243: Does a TBR of 0.5 mean that the fluorophore is better accumulated in the healthy tissue than in the tumor? Even for a tumor lacking the target antigen such a result is hard to understand. Please, comment.

Although the MiaPCACA2 cell line is used as a negative control, it still expresses low levels of CEA which leads to some accumulation of the aCEA nanobody probe. The background values for the skin are comparable between the high and low CEA expressing cell lines (Lwin et al Figure 4C). The manuscript has been modified to reflect this comment. ” This TBR of 0.5 in MiaPACA2 pancreatic tumors is attributable to the low level of CEA expressed in the cell line.”

  1. P. 7, l. 283: A typing error ”to evaluate”.

This has been corrected in the manuscript: ”clinically available devices to evaluate”

  1. P. 7, l. 297: Another possibility to prolong the circulation time of the probes is to use a synthetic hydrophilic polymer as a carrier of the fluorophore and the targeting ligand. Several relevant papers should be cited here (e.g. Multifunct. Mater. 2 (2019) 024004; Pharmaceutics 2020, 12, 31; Theranostics 2020; 10(4): 1948-1959

The discussion on possible use of HPMA as a molecule to prolong serum half-life has been added to the manuscript with the interesting suggested manuscripts. ” Other synthetic molecules such N-(2-Hydroxypropyl) methacrylamide (HPMA), a hydrophilic polymer can increase circulation time83. This has been shown to be efficacious in peptide-based probes against EGFR which face similar challenges as nanobodies in clearance84–86. The approaches permit titration of serum half-life to increase the window of time for probe and tumor binding.”

Sincerely,

Thinzar Lwin, MD MS

Robert Hoffman, PhD

Michael Bouvet, MD

Reviewer 2 Report

Review: Unique benefits of tumour-specific nanobodies for fluorescence-guided surgery by

Thinzar M. Lwin et al.

This review describes the state-of-the-art of tumor imaging with fluorescently labeled nanobodies. The testing of nanobodies for imaging purposes is still in the pre-clinical phase.

The paper focuses on the fluorescent guided surgery (FGS), but surgical aspects were poorly described herein. For example, the description of the success rate of tracking the primary tumor, sentinel node and further metastasis is missing. Secondly, what is the success of removing tumor tissues, i.e. the amount of tumor tissue that remained in the animal after surgery?

In the discussion section, the optimizing of imaging has been proposed. In what respect is the amount of nanobody accumulating in the tumor and for acquiring similar results, what is the estimated dose in patients to deliver sufficient signal to the tumor? Describe the uptake and limitations of nanobodies accumulation in non-target tissue in the vicinity of tumor tissue.

Various fluorescent dyes on nanobodies were described. Are these allowed for human use regarding cytotoxicity and non-specific uptake in non-target tissue?

Table 1 is missing the legend. Also, describe that these all contain pre-clinical studies.

The production of nanobodies in high yield cell cultures (yeasts, bacteria) harbors the risk of contamination with LPS contamination. Describe this issue in the paper. Can nanobodies be produced in mammalian cells as well which sound safer? Sometimes, nanobodies carry HIS-tags, which can be useful for easy purification from the culture. Is this also used for the nanobodies, described in the review?

Add a paragraph for using hybrid nanobodies (i.e. radioactive and fluorescent nanobodies) for tumor cell tracking.

Author Response

February 1, 2021

Dr. Vladimir N. Uversky, PhD, DSc

Editor-in-Chief, Biomolecules

Dear Dr. Uversky and reviewers,

Thank you very much for your careful consideration of our manuscript for publication in Biomolecules. We respect the feedback from the reviewers on this paper and appreciate the opportunity to strengthen specific aspects of this manuscript. In our revised manuscript, we have addressed the comments made by the reviewers point by point, and believe that our responses will satisfy their concerns.

All changes to the document are highlighted in yellow on the revised manuscript. Point-by-point responses (in italics) are given below.

This review describes the state-of-the-art of tumor imaging with fluorescently labeled nanobodies. The testing of nanobodies for imaging purposes is still in the pre-clinical phase.

The paper focuses on the fluorescent guided surgery (FGS), but surgical aspects were poorly described herein. For example, the description of the success rate of tracking the primary tumor, sentinel node and further metastasis is missing. Secondly, what is the success of removing tumor tissues, i.e. the amount of tumor tissue that remained in the animal aft`qer surgery?

The oncologic impact of FGS on the tumor is very heterogenous as this is currently a developing field. The sensitivity, specificity of the molecule and its accuracy for detection of the primary tumor, nodes, and metastases is still being studied at this time. Direct comparisons between bright light and fluorescence guided surgery in animal models have shown superiority using contrast enhancement via fluorescence, but similar data from clinical trials is not yet available even with antibody based probes.

In the discussion section, the optimizing of imaging has been proposed. In what respect is the amount of nanobody accumulating in the tumor and for acquiring similar results, what is the estimated dose in patients to deliver sufficient signal to the tumor?

Oliviera et al demonstrated that “there was up to 17% of the injected dose per gram of tumor (ID/g) 2 hours after injection of the nanobody while there was up to 10% ID/g after injection of cetuximab.” It is unclear the estimated dose in patients and further work will need to be performed to better delineate this, beyond simply scaling up based on weight.  

Describe the uptake and limitations of nanobodies accumulation in non-target tissue in the vicinity of tumor tissue.

Renal elimination of the probe due to the small size of the molecule can also lead to off target signal. As this is a known issue, nanobody based molecules cannot be used for kidney tumors. There may also be limitations on tumors with anatomic locations adjacent to the kidney, the kidney will need to be mobilized to avoid shine through of the signal.

Various fluorescent dyes on nanobodies were described. Are these allowed for human use regarding cytotoxicity and non-specific uptake in non-target tissue?

A discussion on clinically available fluorescent dyes has been added to the manuscript. ” Targeting probes in clinical trials have been conjugated to IRDye800CW since the dye is readily available and have been demonstrated to show safety and efficacy in Phase I/II clinical trials30,31. Anti-VEGF antibodies (bevacizumab-800CW30) and anti-EGFR antibodies tagged to IRDye800CW (panitumumab-IRDye800CW and cetuximab-IRDye800CW31) are such examples.” Also added is a comment on zwitterion dyes: “Neutral or zwitterion based fluorophores, in Phase I clinical trials… “

However clinical work with LICOR’s IRDye700 is still pre-clinical. SGM-101, an anti-CEA antibody conjugated to a 700nm dye is a proprietary fluophore developed by the group called BM104. “Common NIR fluorophores and their status in clinical use are well summarized in Hong et al Table 127

Table 1 is missing the legend. Also, describe that these all contain pre-clinical studies.

The manuscript has been corrected to reflect this. ”Table 1. A summary of pre-clinical nanobody probes being explored for tumor-specific FGS.”

The production of nanobodies in high yield cell cultures (yeasts, bacteria) harbors the risk of contamination with LPS contamination. Describe this issue in the paper. Can nanobodies be produced in mammalian cells as well which sound safer?

Nanobodies have not had issues with LPS contimniation. While it is possible to be produced in mammalian cells, the higher cost of production and demonstrated safety in E Coli production have led to continued use of fermentation for production of the molecules. The manuscript has been modified to reflect this discussion. ”Clinical trials using nanobodies as either therapeutic agents or as a radio-tracer have shown no evidence of immunogenicity54–57. The molecules were produced in E Coli without adverse reactions from lipopolysaccharide contamination and the reported endotoxin level was found to be 0.01 EU/mg58. Caplacizumab, an anti-von Willebrand factor (vWF) nanobody, developed for the treatment of thrombotic thrombocytopenic purpura (TTP) is the most developed nanobody, having completed phase III clinical tri-als56,59. In the field of tumor imaging, anti-Her2 nanobodies conjugated to radiotracers (I-131 and Ga-68) to image patients with breast cancer have completed phase I clinical trials and are undergoing phase II clinical trials55,60. A summary of nanobodies current-ly under clinical trials is well summarized in Jovčevska et al Table 148.”

Sometimes, nanobodies carry HIS-tags, which can be useful for easy purification from the culture. Is this also used for the nanobodies, described in the review?

Nanobodies can be purified without Hex-HIS tags although this is the easiest approach and used for these pre-clinical studies. Other approaches used to purify nanobodies include cation exchange resin or introducing site directed mutagenesis to improve binding to Staphylococcus aureus Protein A (SpA). For Caplacizumab, a cation exchange resin based on its high isoelectric point.

Add a paragraph for using hybrid nanobodies (i.e. radioactive and fluorescent nanobodies) for tumor cell tracking.

The manuscript has been modified to reflect this. ” Besides photosensitizers, hybrid nanobodies can created by conjugation with radiolabels for tumor-specific PET imaging as well as therapeutic radionuclides103. It is possible to consider the use of one molecule for both pre-operative tumor specific imaging and active intra-operative navigation. Nanobody can also carry therapeutic drugs or pro-drugs as well as nanoparticles49,104. However due to renal filtration of small molecules such a nanobodies, renal toxicity can be a potential hurdle.Nanobody technology can be a possible avenue to combine tumor-specific optical fluorescence imaging with drugs or radionuclides for a multi-modal approach to treat cancer.”

Sincerely,

Thinzar Lwin, MD MS

Robert Hoffman, PhD

Michael Bouvet, MD

Reviewer 3 Report

The review by Lwin TM et al. is a convincing presentation of the potential contribution of nanobodies to the field of FGS.

Globally, the field of FGS using antibodies is well presented thanks to a presentation through the target antigens. The authors could have stressed more on the only compound currently in phase III (SGM-101) by citing the corresponding clinical trials publications (Boogerd LSF et al. Lancet Gastroenterol Hepatol. 2018 ; 3:181–191; Hoogstins CES et al. Ann Surg Oncol 2018. 25:3350-3357).

Although I agree with the fact that nanobodies are an interesting platform to be studied in the FGS context, we should wait results of clinical trials to be sure. PK in mice are quite different from PK in Human. Thus, it remain to be demonstrated that nanobodies based FGS compounds will be efficient for FGS. This should be more discussed in the manuscript.

The delay between injection and surgery is an important factor for the acceptance and the routine clinical development of FGS. Injecting FGS compounds 2 to 4 days before surgery might seem difficult but it offers a large imaging window. In such a case, no problem if the surgery is delayed by some hours. With a very short time to TBR peak, any delay in the surgery could have a disastrous impact on FGS results. A sentence discussing this point should be added.

Looking for all the published clinical trials using antibody-dye conjugates, one will see that the molar dye-to-mab ratio is always between 1 and 2 for a carrier molecule (MAb) of 150 000 kDa. It means that 1 dye molecule will be a maximum (if not too much) for nanobodies. A sentence discussing this point should be added.

Author Response

February 1, 2021

Dr. Vladimir N. Uversky, PhD, DSc

Editor-in-Chief, Biomolecules

Dear Dr. Uversky and reviewers,

Thank you very much for your careful consideration of our manuscript for publication in Biomolecules. We respect the feedback from the reviewers on this paper and appreciate the opportunity to strengthen specific aspects of this manuscript. In our revised manuscript, we have addressed the comments made by the reviewers point by point, and believe that our responses will satisfy their concerns.

All changes to the document are highlighted in yellow on the revised manuscript. Point-by-point responses (in italics) are given below.

The review by Lwin TM et al. is a convincing presentation of the potential contribution of nanobodies to the field of FGS.

Globally, the field of FGS using antibodies is well presented thanks to a presentation through the target antigens. The authors could have stressed more on the only compound currently in phase III (SGM-101) by citing the corresponding clinical trials publications (Boogerd LSF et al. Lancet Gastroenterol Hepatol. 2018 ; 3:181–191; Hoogstins CES et al. Ann Surg Oncol 2018. 25:3350-3357).

The manuscript has been corrected to reflect this. ” The manuscript has been corrected to reflect this” Fluorescent anti-VEGF and EGFR antibodies are currently undergoing phase I and II clinical trials29–32. Fluorescent anti-CEA antibody (SGM-101) is currently undergoing phase III clinical trial33,34

Although I agree with the fact that nanobodies are an interesting platform to be studied in the FGS context, we should wait results of clinical trials to be sure. PK in mice are quite different from PK in Human. Thus, it remain to be demonstrated that nanobodies based FGS compounds will be efficient for FGS. This should be more discussed in the manuscript.

A section on the manuscript has been added to reflect this point. “While the details of translation of pharmacokinetics from murine models to humans may need to be further evaluated, the time frame is promising. Caplacizumab, the anti-vWF nanobody, was shown to have a maximal serum concentration of 20% 6-7 hours after subcutaneous administration57. This shorter time to imaging is promising for clinical translation as patients are not required a separate clinical visit 1-2 days before surgery for probe administration and washout.”

The delay between injection and surgery is an important factor for the acceptance and the routine clinical development of FGS. Injecting FGS compounds 2 to 4 days before surgery might seem difficult but it offers a large imaging window. In such a case, no problem if the surgery is delayed by some hours. With a very short time to TBR peak, any delay in the surgery could have a disastrous impact on FGS results. A sentence discussing this point should be added.

Nanobodies still retain a tumor-specific fluorescence signal (although not optimal with the highest contrast) past several hours. The manuscript has been modified to reflect this discussion point. ”While nanobodies can bind deep pockets with nanomolar affinities, the rapid renal elimination of nanobodies leads to a very short serum half-life and a smaller window for exposure and binding to antigen. This can lead to concerns over non-optimal imag-ing should there be any delays in surgery. However in the pre-clinical studies cited in this review that look at the fluorescence signal over time shows that there is still tumor specific fluorescence at up to 72 hours with TBR’s above 1.517. As the molecules have rapid accumulation, there is also the potential for repeat dosing of the probe although there remains work to be done in this area to optimize this strategy.”

Looking for all the published clinical trials using antibody-dye conjugates, one will see that the molar dye-to-mab ratio is always between 1 and 2 for a carrier molecule (MAb) of 150 000 kDa. It means that 1 dye molecule will be a maximum (if not too much) for nanobodies. A sentence discussing this point should be added.

The manuscript has been modified to reflect this discussion point. "Nanobodies are small molecules and can realistically bear only one fluorophore conjugate per nanobody..."

Thank you for the careful consideration of our manuscript.

Sincerely,

Thinzar Lwin, MD MS

Robert Hoffman, PhD

Michael Bouvet, MD

Round 2

Reviewer 1 Report

I recommend to accept the revised version of the manuscript for publication in Biomolecules. The following spelling errors should be corrected:

p. 2, l. 56 "extravaste"

p. 5, l. 147: "Cystiene"

p. 8, l. 340: "N-(2-Hydroxypropyl)" should be without capital H

Author Response

Thank you very much for the constructive criticism of our manuscript for publication in Biomolecules. The revised manuscript has been updated to reflect the recommend changes.

p. 2, l. 56 "extravaste" => "It is thought that the molecules extravasate from leaky tumor vessels..."

p. 5, l. 147: "Cystiene" => "Cysteine-malemide conjugation involves the introduction of a cysteine residue on the surface of the molecule ..."

p. 8, l. 340: "N-(2-Hydroxypropyl)" should be without capital H => "Other synthetic molecules such as N-(2-hydroxypropyl) methacrylamide (HPMA)..."